# Inferring Drug Set and Identifying the Mechanism of Drugs for PC3

**DOI:** 10.3390/ijms25020765

**Published:** 2024-01-07

**Authors:** Shinuk Kim

**Affiliations:** College of Engineering, Sangmyung University, Cheonan 31066, Republic of Korea; kshinuk@smu.ac.kr; Tel.: +82-41-550-5452

**Keywords:** drug interaction, feature selection, NMF

## Abstract

Drug repurposing is a strategy for discovering new applications of existing drugs for use in various diseases. Despite the use of structured networks in drug research, it is still unclear how drugs interact with one another or with genes. Prostate adenocarcinoma is the second leading cause of cancer mortality in the United States, with an estimated incidence of 288,300 new cases and 34,700 deaths in 2023. In our study, we used integrative information from genes, pathways, and drugs for machine learning methods such as clustering, feature selection, and enrichment pathway analysis. We investigated how drugs affect drugs and how drugs affect genes in human pancreatic cancer cell lines that were derived from bone metastases of grade IV prostate cancer. Finally, we identified significant drug interactions within or between clusters, such as estradiol-rosiglitazone, estradiol-diclofenac, troglitazone-rosiglitazone, celecoxib-rofecoxib, celecoxib-diclofenac, and sodium phenylbutyrate-valproic acid.

## 1. Introduction

Prostate cancer (PC) is the second most common cancer in men and the fourth most common cancer overall. As of 2020, there were more than 1.4 million new cases of prostate cancer in the world [1]. In addition, prostate cancer, known as being complex and heterogeneous, is the second leading cause of death among men in the United States. Incidences of prostate cancer decreased rapidly from 2007 to 2014 due to screening recommendations such as prostate-specific antigen. However, the incidence rate has increased by 3% per year since 2014. According to cancer statistics from the American Cancer Society, 288,300 new cases and 34,700 deaths were reported for 2023.

Drug repositioning is a novel strategy for discovering new applications of existing drugs for use in various diseases. It entails using existing medications to treat conditions other than those for which they were originally designed. As a result, drug repositioning is important in optimizing the preclinical process of developing novel drugs. Computational drug repurposing can significantly reduce drug development costs and time by discovering new applications for existing drugs. Despite recent advances in the field of computational drug repositioning, however, developing robust models remains a complex process fraught with difficulties.

One promising strategy for drug repositioning is a computational method based on gene expression analysis that uses drug-induced disease datasets, chemical–chemical networks, and protein–protein networks to identify target genes in a specific disease. Such repositioning techniques have been successful in obtaining drug approval for conditions other than the original target. However, most potential drugs fail the validation process. Even if a drug target can be identified using genetic pathway data, the efficacy of the drug in treating diseases is unknown [2].

Recently, Ma et al. [3] published a paper in which they described a method for repositioning pancreatic cancer drugs based on similarities in gene expression patterns. According to that study, the fundamental idea behind computational drug repositioning is that genes with similar patterns of expression may respond to drugs with similar therapeutic efficacies. Previous research on the anatomical therapeutic chemical classification of drugs has focused on drug–drug interactions and drug–target gene correlations [4,5]. 

Before conducting laboratory studies, it is necessary to select and prioritize candidate genes for specific diseases because identifying disease genes from large candidate genes in experimental settings is a very expensive and time-consuming task. There are numerous methods for prioritizing genes based on machine learning. These methods differ in several ways, including the feature vectors of genes used, the datasets used, which have different structures, and the learning model [6]. Human prostate cancer PC-3 cell lines are thought to represent late stages of prostate cancer and have been extensively used to study prostate cancer progression and develop therapeutic agents [7]. In this paper, using datasets from drug-induced prostate adenocarcinoma PC-3 cell lines that were derived from bone metastases of grade IV prostate cancer, we explore the interactions between drugs and genes by identifying differential gene expression with two feature selection approaches. 

## 2. Results

### 2.1. Inferring Pathways by Drug Clusters Using All Features

We clustered drug-induced tumor datasets consisting of 22,283 genes and 44 PC3 samples. The results of the NMF performance from rank 2 to 5 are shown in Figure 1a–e. In the results, rank 2 performs the best accuracy of 0.985 in Table 1, where cluster 1 contains 27 samples and cluster 2 contains 17. The robustness of the NMF performance was confirmed by using the k-means clustering method, in which the silhouette score of 0.36 was found to be the best at rank 2, compared with 0.27 at rank 3 and rank 4 [8,9].

In the case of rank 2 clusters, we used GSEA to investigate differences in characteristics between two clusters. Interestingly, when compared to previous clinical studies [10], the enriched pathways in cluster 1 are consistent with pathways associated with a poor prognosis. In contrast, the enriched pathways in cluster 2 are consistent with pathways associated with a good prognosis (Table 2).

Using the DrugBank and STITCH databases, we found interacting drugs within and between clusters, as well as common genes as drug targets. Estradiol is an estrogenic steroid that is used to treat prostate cancer. Estradiol, which belongs to cluster 2, interacts with rosiglitazone (common gene; RETN), valproic acid (common genes; CYP19A1, ESR1, ESR2, and PTGS1), diclofenac (common genes; PTGS2 and CYP2A6), tretinoin (common genes; RAPA, AR, ESR1, ESR2, and HSD17B1) in cluster 1 and with genistein (common genes; AR, ESR1, ESR2, SHBG, and CYP19A1), fisetin (common gene; AR), resveratrol (common genes; ESR1, ESR2, and CYP19A1), and fulvestrant (common genes; AR, ESR1, and ESR2) in cluster 2. Valproic acid interacts with diclofenac (common genes; ABCB1, CYP3A4, and CYP2C19, CYP2C9), tretinoin, and rosiglitazone (common gene; LEP) within cluster 1. Tretinoin interacts with diclofenac (common genes; PTSG1, PTGS2, and PPARG) and rosiglitazone (common genes; PTGS2, PPARG, and RXRA) within cluster 1. Celecoxib interacts with diclofenac (common genes; PTGS1, PTGS2, CYP2C9, and VEGFA) within cluster 1. Troglitazone and rosiglitazone interact with the common genes PPARA, RXRA, UCP2, RETN, LEP, and CD36. Celecoxib, rofecoxib, and indomethacin all interact with the common genes PTGS1 and PTGS2. Indormethan, faudil, copper sulfate, TTNPB, and trichostain A do not interact with one another.

When comparing pathways between entrances A and B, as shown in Table 2, common pathways account for 37.5% of the total in cluster 1. As shown in Figure 1a–d, clusters 1 in rank 2 and rank 3 are similar, but cluster 3 in rank 3 is derived from cluster 2 in rank 2. In total, three pathways in clusters A and B overlap, as do three pathways in clusters 2 and 3 in rank 2.

In the case of rank 3, cluster 1 contains 21 distinct drugs, cluster 2 contains 5 drugs, and cluster 3 contains 17 drugs. According to the Cophenetic coefficient, rank 3 is well separated between clusters, and the drugs do not overlap with each other except for valproic acid, which appears in both cluster 1 and cluster 2. The drugs in cluster 2 from rank 3 (Appendix A) are 4,5-dianilinophthalimide (DAPH, DB12362), deferoxamine (DB00746), fisetin (DB07795), genistein (DB01645), and valproic acid (DB00313). Among them, the DrugBank interaction showed that valproic acid and genistein interact with one another in such a way that genistein metabolism is somewhat suppressed when combined with valproic acid [11]. Similarly, the drugs in Cluster 3 mercaptomurine-estradiol interact with one another [11]. 

The enrichment pathways were unable to identify overlapping genes (which could indicate effected genes from the set of drugs) among the pathways. However, we used statistical tests to identify the interesting genes. Using significance analysis of microarray (SAM), we discovered the top three differentially expressed genes between two clusters from rank 2, and they were FCGRT, NBR2, LHB over-expressed in cluster 2, SMA4K, ZMAT4, and CIDEA over-expressed in cluster 1. The results indicate the most affected genes by drugs to be within or between clusters, despite the fact that those genes did not appear in the common genes for drug interaction used in this study.

### 2.2. Drug Interaction with Clustering Using Feature Selection

We classified the drugs based on genes that were significantly different in normal and cancer samples. We first identified features that were differentially expressed for the strategy using the fsv and Fisher features approaches, yielding 30 genes with each method. Because the selection criteria for each feature method differ, only four overlapping genes, RPLP0, RPL11, RPS18, and LOC642741 (also known as RPL3P7), were identified. Interestingly, as shown in Table 3, the overlapping genes are all members of the ribosome protein family.

For ranks 2–4, we combined the two feature selection methods fsv and Fisher with NMF clustering methods. First, we chose to use 30 and 20 genes with each method. Then, we calculated the Cophenetic coefficients by performing NMF, and they are shown in Figure 2a,b. The clustering method with 30 features chosen from fsv produced a better separation than the Fisher method with 30 features. When compared to the results obtained using all genes, the 20 genes chosen from the fsv method within rank 2 produced the best results.

As shown in Table 4, NMF clustering using 20 and 30 genes selected from fsv revealed six and thirty-one overlapping drugs among the drugs included in clusters 1 and 2, respectively.

We used the chemical interaction in STITCH [12] and DrugBank (https://go.drugbank.com/ accessed on 10 October 2023) [11] to obtain the drug interaction. In cluster 2, celecoxib is related to rofecoxib, estradiol, imatinib, rosiglitazone, valproic acid, alpha-estradiol, and diclofenac. Except for estradiol and alpha-estradiol, the genes are also found in cluster 1 when all features are used. Furthermore, we discovered more drug interactions in cluster 2 where drug information is available (details in Appendix A). The STITCH database revealed that celecoxib interacts with indomethacin (common genes; PTGS1, TPGS2, and IL6) and docosahexaenoic (common genes; PTGS1, PTGS2, and MPO) within cluster 1. 

## 3. Discussion

This study proposed a novel computational method for identifying drug interactions in PC3 cell lines based on drug-induced gene expression profiles. We began by clustering using all feature methods. Among the clusters, estradiol interacts with rosiglitazone, diclofenac, tretinoin, and genistein. Within cluster 1, valproic acid interacts with diclofenac, tretinoin, and rosiglitazone.

According to the STITCH database, troglitazone and rosiglitazone interact with PPARA, RXRA, UCP2, RETN, LEP, and CD36 common genes. Valproic acid and rosiglitazone share only one gene, LEP. Celecoxib, rofecoxib, and indomethacin all interact with the PTGS1 and PTGS2 common genes. Indormethan, faudil, copper sulfate, TTNPB, and trichostain A either do not interact or do not appear in the STITCH database.

Next, we selected candidate genes from tumor and normal datasets using the fsv method and clustered drug-induced samples using the tumor expression datasets of those genes only. As a result, we discovered that celecoxib in cluster 1 and rofecoxib in cluster 2 share the PTGS1 and PTGS2 genes, implying a connection. DDI based on the chemical similarity of the DrugBank is used to cross-check the candidate drug interactions we discovered.

Our findings based on gene expression are the results of treatment, and chemical similarities are the static datasets. Rofecoxib has been used for the treatment of many diseases, such as osteoarthritis, rheumatoid arthritis, acute pain, primary dysmenorrhea, and migraine (https://go.drugbank.com/drugs/DB00533, accessed on 10 October 2023) [13,14]. Rofecoxib also has been considered to treat human prostate cancer [15,16]. Celecoxib in cluster 1 and diclofenac in cluster 2 share genes for PTGS1, PTGS2, CYP2C9, and VEGFA according to the STITCH database. Celecoxib was previously used to inhibit human prostate cancer [17]. Diclofenac was used as a non-steroidal anti-inflammatory drug to treat osteoarthritis and rheumatoid arthritis [18,19] and to inhibit prostate cancer tumor growth [20]. When all genes are used, however, celecoxib, rofecoxib, and diclofenac are all included in the same cluster. Furthermore, in cluster 2, alpha-estradiol is related to estradiol (DB00783) via the shared target genes ESR1, ESR2, CYP19A1, and PRL, whereas sodium phenylbutyrate (4-phenylbutyra) is related to valproic acid (DB00313, Valproate) via no common genes. Estradiol has been used to slow the growth of prostate cancer [21]. Estradiol has repeatedly demonstrated its efficacy as an estrogen steroid, including its multifaceted mechanism [22], local synthesis in the newborn brain [23], and use as a neuroprotective agent [24].

## 4. Materials and Methods

The microarray profiles (accession number GSE5258-GPL96) were obtained from the gene expression omnibus (GEO; https://www.ncbi.nlm.nih.gov/geo, accessed on 10 October 2023) [25], a database repository of high throughput gene expression data and hybridization arrays, chips, and microarrays. In brief, the datasets were updated in December 2017 with 346 cell lines containing HL60, MCF7, PC3, SKMEL5, and ssMCF7. With 22,283 probes, 111 distinct drugs are induced in those cell lines. From the datasets, we extracted 56 PC3 cell lines consisting of 12 normal and 44 prostate cancer samples, and the cancer samples were induced by 38 unique drugs. The 22,283 probe IDs were converted to 9357 unique gene symbols using the GPL 96 platform (Affymetrix human genome U133A array).

As a clustering method, we adopted the non-negative matrix factorization method (NMF), which is a mathematical and computational technique used in gene expression data analysis. It is intended to factorize a given non-negative data matrix into two or more lower-dimensional non-negative matrices. NMF has gained popularity due to its ability to uncover latent structures, patterns, and features in data, particularly when the data is inherently additive such as gene expression data. The NMF algorithm uses factorizing gene expression profiles based on positive matrix decomposition [26,27].

The main concept is presented as a matrix of n (genes) x m (samples) decomposed by W and H; A∼WH, where W is a n×k matrix, and H is a k×m matrix, with k as the number of clusters. The cost function is:$$D(A||WH)=\sum_{ij}\left(A_{ij}\log\frac{A_{ij}}{{\left(WH\right)}_{ij}}-A_{ij}+{\left(WH\right)}_{ij}\right).$$

For each iteration, the updating criteria for components of matrix W and H are:$$W_{ia}\leftarrow W_{ia}\frac{\sum_{u}H_{au}A_{iu}/{\left(WH\right)}_{iu}}{\sum_{v}H_{av}},\;H_{au}\leftarrow H_{au}\frac{\sum_{i}W_{ia}A_{iu}/{\left(WH\right)}_{iu}}{\sum_{k}W_{ka}}.$$

To determine the optimal number of clusters, we calculated Cophenetic coefficients according to the following equation [10,28]:$$c=\frac{\sum_{i<j}\left(Y_{ij}-\gamma )(Z_{ij}-z\right)}{\sqrt{\sum_{i<j}{\left(Y_{ij}-\gamma \right)}^{2}\sum_{i<j}{\left(Z_{ij}-z\right)}^{2}}}$$
where $Y_{ij}$ represents the distance between i and j, and $Z_{ij}$ represents the dendrogrammatic distance between i and j. Higher cophenetic correlation coefficients indicate superior clusters.

Using the clusters as groups, we performed gene set enrichment analysis (GSEA) to identify enriched pathways [29]. There are many sources of feature selection methods available, including significance analysis of microarray (SAM) from R (https://www.R-project.org/, accessed on 10 October 2023) [30] and the MATLAB feature selection library (https://www.mathworks.com, accessed on 10 October 2023) [31]. Among them, we adopted the Fisher feature selection and the feature selection concave (fsv) methods as a tool for identifying differentially expressed gene sets. The fsv method was chosen to approach the global minimum because optimization methods frequently become stuck in a local minimum [32]. 

The Fisher feature selection method, which is a dimension reduction technique in datasets [32], computes a discriminate score for two groups A,B based on the formula ${\left(\mu_{A}-\mu_{B}\right)}^{2}/(\sigma_{A}^{2}+\sigma_{B}^{2})$, where μ’s are means and σ’s are standard deviations [11,12].

Drug–drug interaction (DDI) information based on chemical similarity to the drug involved was posited using the combination score from STITCH (http://stitch.embl.de/cgi/input.pl, accessed on 10 October 2023) [33] and DrugBank database (https://go.drugbank.com/, accessed on 10 October 2023) [13,34]. We employed the DrugBank’s drug interaction checker, in which drug–drug interactions can be sorted into two broad categories: pharmacokinetic drug interactions and pharmacodynamic drug interactions. Pharmacokinetic drug interactions occur when one drug affects the other through the body, whereas pharmacodynamic drug interactions occur when one drug alters the actual clinical effect of the other. Although there is an issue about the reliability between DDI and chemical similarity, the DrugBank’s drug interaction checker is widely used due to its ease of use and quick results. The overall flow chart of the method is shown in Figure 3.

To differentiate tumor samples induced by specific drugs using NMF, we used feature selection methods to identify the most differentially expressed genes between normal and tumor samples. After obtaining the genes, we extracted datasets from tumor samples that included drug information.

## 5. Conclusions

In this study, we proposed a computational method to investigate the efficacy of systemic drug studies for prostate cancer. First, we grouped tumor samples using NMF and found the enriched pathways according to clusters. With respect to the clusters 1 and 2, our findings about the enriched pathways are consistent with previous research in terms of poor and good prognosis [10]. Second, we used feature selection techniques (fsv, Fisher) and clustering techniques to describe drug interactions based on gene expression profiles (NMF) for prostate cancer. Rank 2 resulted in the strongest cluster, implying that the activities of the drugs and the target genes are located within the cluster.

We identified the following drug interactions between or within clusters: celecoxib-diclofenac with common genes such as PTGS1, PTGS2, CYP2C9, and VEGFA; celecoxib-rofecoxid with common genes such PTGS1 and PTGS2; alpha estradiol-estradiol with common genes ERS1, ERS2, CYP19A1, and PRL; and sodium phenylbutyrate-valproic acid interaction within cluster 1 using all features and between clusters using feature selection. However, the clustering methods are sensitive to the size of the features. Even though we combined information to determine drug interactions for drug repositioning, the related drugs are not consistently clustered together. Consequently, clinical databases related to drug-induced events need to be incorporated in a future study. In addition, although the DDI methodology based on chemical similarity has been widely used, its reliability is still unclear, and it should be resolved in future work.

## Figures and Tables

**Figure 1 ijms-25-00765-f001:**
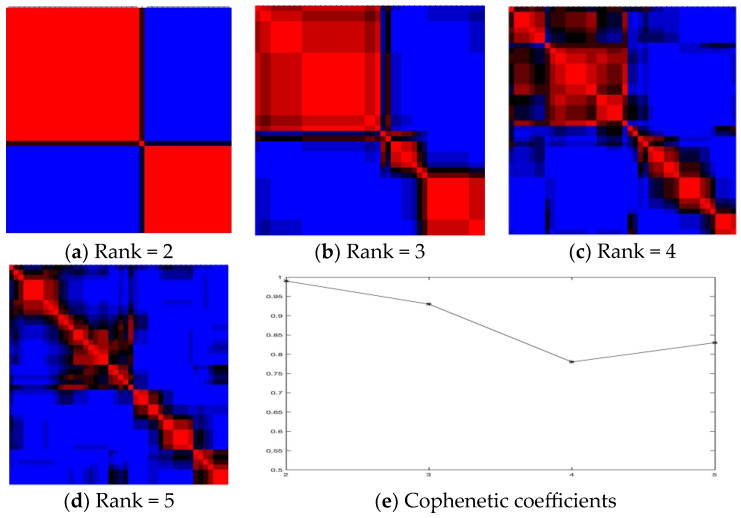
(**a**) Rank 2, Cophenetic coefficient = 0.985; (**b**) Rank 3, Cophenetic coefficient = 0.933; (**c**) Rank 4, Cophenetic coefficient = 0.786; (**d**) Rank 5, Cophenetic coefficient = 0.838; (**e**) Ranks 2–5, *x*-axis represents cluster, and *y*-axis represents Cophenetic coefficients.

**Figure 2 ijms-25-00765-f002:**
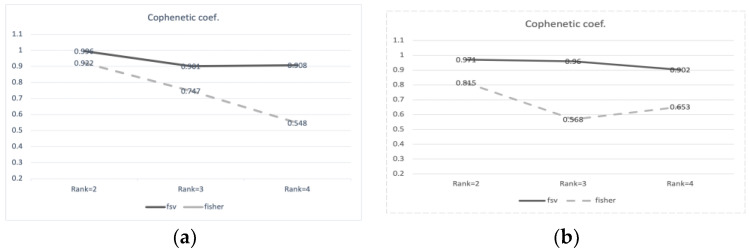
Cophenetic correlation coefficients ranks 2 to 4. (**a**) Number of selected features is 20; (**b**) number of selected features is 30.

**Figure 3 ijms-25-00765-f003:**
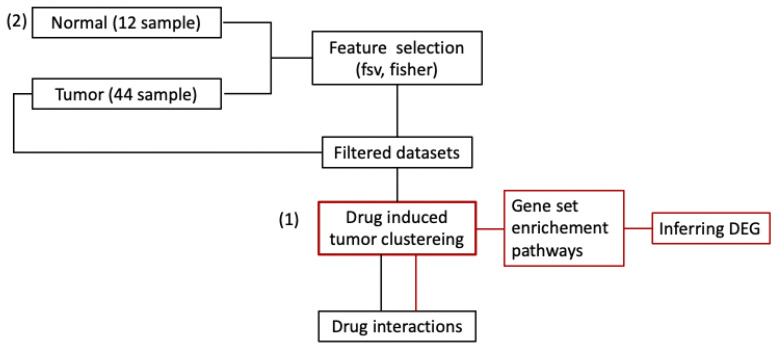
The overall flow chart of the methods. (1) Red lines represent inferring differentially expressed genes with all genes; (2) black lines represent inferring drug interactions with feature selection.

**Table 1 ijms-25-00765-t001:** Clustering by drugs; unique drugs are presented.

Drugs in Cluster 1	Drugs in Cluster 2
LY-294002	fulvestrant
Rosiglitazone	genistein
Troglitazone	alpha-estradiol
17-allylamino-geldanamycin	monastrol
Valproic acid	mercaptopurine
Sodium phenylbutyrate	butirosin
Novobiocin	estradiol
Fasudil	docosahexaenoicacidethylester
Diclofenac	U0125
15-deltaprostaglandin J2	resveratrol
Tretinoin	splitomicin
Trichostatin A	dimethyloxalylglycine
Monorden	HNMPA-(AM)3
TTNPB	butein
indomethacin	fisetin
Tetraethylenepentamine	4,5-dianilinophthalimide
Rofecoxib	deferoxamine
Copper sulfate	
Celecoxib	
Imatinib	
Pirinixic acid	

**Table 2 ijms-25-00765-t002:** Enrichment pathways in two clusters (**A**) and enrichment pathways in three clusters (**B**).

**(A)**		
**Cluster 1**	**Cluster 2**	
HSA00531_GLYCOSAMINOGLYCAN_DEGRADATION	HSA00040_PENTOSE_AND_GLUCURONATE_INTERCONVERSIONS	
HSA00910_NITROGEN_METABOLISM	HSA03050_PROTEASOME	
HSA04950_MATURITY_ONSET_DIABETES_OF_THE_YOUNG	HSA03010_RIBOSOME	
HSA00565_ETHER_LIPID_METABOLISM	HSA03060_PROTEIN_EXPORT	
HSA04080_NEUROACTIVE_LIGAND_RECEPTOR_INTERACTION	HSA00680_METHANE_METABOLISM	
HSA04742_TASTE_TRANSDUCTION	HSA00062_FATTY_ACID_ELONGATION_IN_MITOCHONDRIA	
**(B)**		
**Cluster 1**	**Cluster 2**	**Cluster 3**
HSA00563_GLYCOSYLPHOSPHATIDYLINOSITOL_ANCHOR_BIOSYNTHESIS	HSA00602_GLYCOSPHINGOLIPID_BIOSYNTHESIS_NEO_LACTOSERIES	HSA03010_RIBOSOME
HSA00531_GLYCOSAMINOGLYCAN_DEGRADATION	HSA04060_CYTOKINE_CYTOKINE_RECEPTOR_INTERACTION	HSA00040_PENTOSE_AND_GLUCURONATE_INTERCONVERSIONS
HSA00910_NITROGEN_METABOLISM	HSA00430_TAURINE_AND_HYPOTAURINE_METABOLISM	HSA03060_PROTEIN_EXPORT
HSA04614_RENIN_ANGIOTENSIN_SYSTEM	HSA04740_OLFACTORY_TRANSDUCTION	HSA03050_PROTEASOME
HSA00565_ETHER_LIPID_METABOLISM	HSA04940_TYPE_I_DIABETES_MELLITUS	HSA00720_REDUCTIVE_CARBOXYLATE_CYCLE

**Table 3 ijms-25-00765-t003:** A list of the selected genes with feature selection via concave (**A**) and Fisher feature selection (**B**) methods. The genes in bold are overlapping.

**(A)**					
**TPT1**	COX6A1	ACTB	ACTB	CCDC72	RPL7
ACTG1(ACTB)	RPS2 (SNORA64)	RPL38	GAPDH	S100A6	EEF1A1
RPS10	HUWE1	GAPDH	**RPS18**	ACTB	FTHP1
**RPLP0**	HSPA1A	ACTG1	ALDOA	**RPL3**	ALDOA
RPL24	**RPL11**	ODC1	UBC	ACTG1	RPS24
**(B)**					
**RPL11**	CCNT1	EIF4H	UBE2L6	SLC36A1	RPL23
MUC6	KLHL24	**RPL3**	TNXA	RPL30	CAMLG
216138_at *	DES	HADHA	ANKRD1	FGF16	POLG2
207756_at *	TBCD	RPL9	NEBL	HTR5A	UBB
**RPS18**	RPL32	217709_at *	LRP2BP	**RPLP0**	EEF1DP5

* No genetic ID.

**Table 4 ijms-25-00765-t004:** The overlapping drugs from NMF clustering using 20 and 30 selected genes.

Overlapping Drugs in Cluster 1	Overlapping Drugs in Cluster 2		
17-allylamino-geldanamycin (2)	LY-294002 (2)	copper sulfate	mercaptopurine
15-delta prostaglandin J2	rosiglitazone	deferoxamine	dimethyloxalylglycine
	troglitazone (2)	rofecoxib	splitomicin
monorden (PGSC0003) (2)	valproic acid (2)	pirinixic acid (2)	alpha-estradiol
celecoxib	sodium phenylbutyrate	monastrol	genistein
	novobiocin	4,5-dianilinophthalimide	fulvestrant
indomethacin	fasudil		resveratrol
Docosahexaenoic acid ethyl ester	diclofenac	fisetin	U0125
	tretinoin	butein	estradiol
	tetraethylenepentamine	HNMPA-(AM)3 TTNPB	butirosin imatinib

## Data Availability

Data can be freely ordered from GEO datasets (https://www.ncbi.nlm.nih.gov/geo/query/acc.cgi?acc=GSE5258/, accessed on 10 October 2023). The access code for the FAIRness is available upon request at kshinuk@gmail.com.

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
