# Peer review of "Inferring Drug Set and Identifying the Mechanism of Drugs for PC3"

_ijms, 2024, doi:10.3390/ijms25020765_

Round 1
Reviewer 1 Report
Comments and Suggestions for Authors
Can be improved
Reviewer 2 Report
Comments and Suggestions for Authors
The purpose of the study conducted and presented cannot be properly deduced from the article. The title, the abstract and the explanation of why the presented study was conducted or the aim of the study indicate prostate cancer as the target disease for which potential new drugs are to be found through repurposing. However, the results are discussed and the conclusions relate to pancreatic cancer.
In the methodology, the transcriptomic datasets used in the study should be presented in more detail. In addition, the scripts used to perform the analysis should be specified.
The results obtained – targets and drugs - should be presented and discussed in relation to prostate cancer (specifically PC3 cell line) or pancreatic cancer to make the results relevant to the treatment of the target cancer.
Due to these serious shortcomings, I cannot recommend the article for acceptance.
Comments on the Quality of English LanguageIt's fine.
Reviewer 3 Report
Comments and Suggestions for Authors
Please see file attached.

The author should thoroughly review the entire manuscript to address typos (e.g.: estridiol instead of estradiol), formatting aspects (especially equations), and improve overall readability.
Round 2
Reviewer 1 Report
Comments and Suggestions for Authors
The authors have addressed the comments.
Author Response
Thanks.
Reviewer 2 Report
Comments and Suggestions for Authors
The author has tried to present the new protocol, but it is not clearly explained why this new protocol is necessary, what advantages it has and why prostate cancer or PC3 cell line are used for its presentation. The materials and methods are not described clearly enough or in sufficient detail. For example, the transcriptomic datasets used in the study are still presented in too general, inappropriate and unclear terms. Furthermore, it is not sufficiently described which data from other analyzed databases are used and how.
The English language should be reviewed throughout the manuscript. For example, „drug-induced prostate adenocarcinoma cell lines" “ and „On those cell lines, 111 unique drugs are induced with 22283 probes“ are not scientifically clear and correct,etc.
The manuscript is not easy to read to recognize its potential and to check the reliability of the results.
Comments on the Quality of English Language
English should be checked.
Author Response
refer to the attached file.

Reviewer 3 Report
Comments and Suggestions for Authors
The author effectively addressed the comments by the reviewers and so, the manuscript in its current form can be published.
Author Response
Thanks.